# NUMBER REPRESENTATIONS IN LLMS: A COMPUTATIONAL PARALLEL TO HUMAN PERCEPTION'

## ABSTRACT

We provide empirical evidence that large language models (LLMs) encode numerical values on a compressed, logarithmic number line, challenging the prevailing assumption of linear representation. Extracting hidden states for numerals, we project them onto one-dimensional manifolds using dimensionality reduction and evaluate two complementary metrics: Spearman's $\rho$ to measure monotonicity and a newly introduced Scaling Rate Index ($\beta$) that quantifies whether spacing is sublinear, linear, or superlinear. Across several LLM families, unsupervised projection into low-dimensional subspaces of maximum variance consistently uncovers strong sublinear trends. Interventions along the discovered number-line directions causally modulate next-number predictions, demonstrating that these dimensions encode genuine numerical structure. This compressed geometry is robustly observed in controlled log-spaced prompts and in real-world settings such as birth years, but is absent in non-numerical controls. Our findings refine the linear representation hypothesis by showing that numerical magnitudes occupy a structured subspace whose internal geometry is systematically non-uniform and logarithmic in nature.

## 1 INTRODUCTION

Do LLMs preserve a uniform spacing of numerical values, or do their representations become increasingly compressed as magnitudes grow? In line with the deep learning tradition of cross-referencing cognitive hypotheses with artificial networks LeCun et al. (2015); Schmidhuber (2015); Hassabis et al. (2017), we investigate the applicability of the *logarithmic mental number line hypothesis*. Rooted in psychophysical studies such as the Fechner–Weber law, this hypothesis formalizes the everyday observation that counting is asymmetric. We as humans list early integers one by one (1, 2, 3, 4, 5), yet describe larger magnitudes in broader categories such as "hundreds," "thousands," or "millions." Its empirical basis is well established in behavioral studies, particularly those showing that young children and individuals without formal mathematics education tend to space numbers logarithmically when asked to place them on a line Fechner (1860); Dehaene (2003); Siegler & Opfer (2003). While formal mathematical training shifts individual perception toward linearity, logarithmic encoding continues to appear in estimation and large-number tasks Dehaene et al. (2008); Moeller et al. (2009).

An influential view, the linear representation hypothesis Park et al. (2023), holds that many concepts in LLMs reside in low-dimensional linear subspaces, making them linearly decodable Heinzerling & Inui (2024). Yet *linear decodability does not imply that values are evenly spaced within those subspaces*. To the contrary, recent probing studies reveal that while order can be recovered with linear probes, precision deteriorates for larger magnitudes, suggesting compressed rather than uniform spacing Zhu et al. (2025). Other analyses show that LLMs represent numbers primarily through their digit sequences rather than as continuous magnitudes. In particular, individual digits can be reconstructed accurately, but the overall number fails to map to a uniform scale Levy & Geva (2024). Moreover, training on higher-base numeral systems impairs extrapolation compared to base-10, reinforcing the view that symbolic digit structure, not continuous magnitude, dominates numerical encoding Zhou et al. (2024). These findings motivate a closer examination of whether the apparent linearity of number representations masks an underlying logarithmic structure (refer to appendix C.1 for further behavioral analysis).

Motivated by this, we test whether LLMs encode numbers with systematic compression. Our approach is twofold. First, we project hidden representations across layers onto one-dimensional

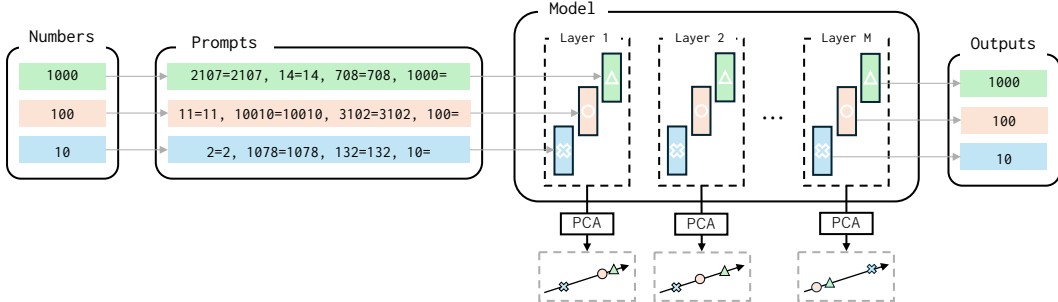

Figure 1: The overall graphical representation of our method. Numbers are passed to the model in form of a prompt and the internal representations are captured from the embeddings corresponding to token '='. At every layer, we perform PCA projections onto one and two dimensional subspaces and pick a layer with highest explained variance ($\sigma^2$) score to further analyze monotonicity and scaling of number representations.

manifolds using Principal Component Analysis (PCA) and Partial Least Squares (PLS). Second, we evaluate two key properties: order preservation, measured by Spearman's $\rho$, and spacing compression, quantified by a Scaling Rate Index ($\beta$) that distinguishes sublinear, linear, and superlinear trends. Unsupervised by labels, PCA provides a geometry-preserving view of hidden states. Solving $\max_{\|w\|=1} \text{Var}(Xw)$, PCA uncovers variance-driven directions that consistently expose sublinear spacing, revealing that numerical magnitudes lie on a compressed number line. By contrast, PLS solves $\max_{\|w\|=1} \text{Cov}(Xw, y)^2$, aligning projections to numeric labels[1]. While this achieves high correlation with $y$, it stretches or compresses inter-number gaps to fit the target, often inflating $\beta$ and obscuring the underlying compression.

Upon deeper scientific analysis of the unsupervised objective, we discover that number representations follow a consistently compressed, logarithmic geometry. We analyze the nature of this subspace by tracing its dimensionality, localizing it to layers of high explained variance, and contrasting it with length-matched non-numeric controls to isolate magnitude from token length. In short, we summarize our contributions as follows:

- We disentangle geometry from decodability, showing that linear decodability can coexist with non-uniform spacing ($\rho, \beta$, ref Table 1). Using one-dimensional PCA projection, we uncover the number line along which we test numerical spacing.

- We introduce the Scaling Rate Index ($\beta$). Computed by means of geometric regression, this novel metric quantifies whether spacing is sublinear, linear, or superlinear. Defined on exponential inputs ($x_i = 10^i$), it complements Spearman's $\rho$ by capturing geometry rather than order, and links $\beta < 1$ to concave number lines (proposition 1).

- We introduce logarithmic compression as a property of LLMs. We observe it consistently across architectures such as LLaMA, Pythia, GPT-2, and Mistral, with effects localizing to specific layers of high explained variance, while non-numeric controls do not exhibit the same structure (table 2).

- We establish the functional relevance of compressed number lines. We show that compression extends beyond synthetic prompts to real-world data such as birth years, but not to population sizes where monotonicity is weak (appendix C.1). Furthermore, interventions along the PCA-discovered number-line direction confirm that these subspaces actively modulate next-number predictions (appendix C.1).

## 2 RELATED WORKS

The subtle distinction between linear decodability and uniform spacing has not been explicitly drawn in prior work, though recent studies provide indirect evidence of it. The linearity of internal

---

[1]Here, $X \in \mathbb{R}^{n \times d}$ is the matrix of $n$ hidden representations (each $d$-dimensional), $y \in \mathbb{R}^n$ the corresponding numeric labels, and $w \in \mathbb{R}^d$ the projection vector.

representations (Park et al., 2023) has been a central assumption in existing research, suggesting that language models encode numerical values in a linear manner. However, this notion of linearity does not imply that values are evenly spaced. Zhu et al. (2025) present an analysis of partial number encoding showing that probing accuracy declines as sequence length increases, with precision deteriorating for larger magnitudes. This pattern is reminiscent of logarithmic encoding, where resolution is higher for smaller numbers. They conclude that LLMs do encode numerical values in their hidden representations, yet linear probes fail to reconstruct these values faithfully. The authors argue that this pattern reflects the use of nonlinear encoding mechanisms in language models. Our findings support this perspective and further reveal the structure of the underlying nonlinearity.

Further evidence for non-uniform geometry comes from studies showing that LLMs represent numbers primarily through base-10 digit sequences rather than as continuous magnitudes (Levy & Geva, 2024). Circular probing in Levy & Geva (2024) reveals that while individual digits are reconstructed accurately, performance declines for larger numbers, pointing to a structured rather than holistic encoding. In addition, Zhou et al. (2024) show that models trained on higher-base numeral systems struggle with extrapolation, implying an implicitly compressed representation where smaller values have finer granularity, consistent with logarithmic scaling. Together, these findings substantiate the view that LLMs encode numbers in a non-uniform, sublinear manner.

The local structure of logarithmic and other non-linear functions is characterized by approximate linearity over small over sufficiently small $\varepsilon$-intervals. As a result, methods such as PLS regression and activation patching (Heinzerling & Inui, 2024; El-Shangiti et al., 2024), which operate on fine-grained activation variations, tend to recover local monotonicity while leaving the broader nonlinear geometry unresolved.

By contrast, global structure is more difficult to assess directly, leading many studies to shift toward numerical reasoning benchmarks that evaluate task-level performance with explicit numbers. For instance, Park et al. (2022) examine unit conversion and range detection, while Zhang et al. (2020) focus on commonsense magnitude comparisons. These tasks provide insight into how LLMs use numbers in context, but they do not probe the spatial organization of number representations within hidden states. Our study instead examines hidden-state geometry directly, distinguishing linear decodability from uniform spacing. By introducing the Scaling Rate Index and contrasting PCA with PLS projections, we quantify the extent to which LLM number representations exhibit globally compressed, logarithmic trends that are systematically embedded across models.

## 3 METHODOLOGY

### 3.1 PROJECTION OF LLM REPRESENTATIONS

The *logarithmic mental number line hypothesis* suggests that humans perceive numerical magnitudes sublinearly. Motivated by this, we ask whether large language models (LLMs) exhibit a similar structure in their hidden representations.

LLMs such as LLaMA-2 process inputs by mapping them into a high-dimensional representation space, where each input $x$ (e.g., a number) is transformed into an internal representation $f(x) \in \mathbb{R}^d$. Examining the geometry of these representations across a set of inputs $\mathcal{X}$ can reveal how the model organizes and reasons about them, for example whether numerical values align along a number line and whether this line exhibits uniform or compressed spacing.

We denote this mapping by $f_{\text{LLM}}$, in analogy to the human perceptual mapping. Our analysis focuses on two properties of $f_{\text{LLM}}$: whether it preserves the natural ordering of numbers, and how it transforms their magnitudes.

To study these properties, we project the hidden states into lower-dimensional subspaces. Specifically, we apply a transformation $T : \mathbb{R}^d \to \mathbb{R}^p$ with $p \in \{1, 2\}$, using Principal Component Analysis (PCA) or Partial Least Squares (PLS). The resulting mapping is

$$f_{\text{LLM}}(x) := T(f(x)), \tag{1}$$

where $f$ denotes the original mapping from an input number to its high-dimensional hidden state.

This projection allows us to analyze the induced geometry directly in one or two dimensions. For any two inputs $x, y \in \mathcal{X}$, we define their distance as $d(x, y) = \|f_{\text{LLM}}(x) - f_{\text{LLM}}(y)\|$, which provides the basis for evaluating monotonicity and scaling behavior in the following sections.

### 3.2 PROPERTY 1: MONOTONICITY

A central question for projected representations is whether numerical order is preserved. If smaller inputs map to smaller coordinates, the projection recovers a meaningful number line. This is equivalent to requiring *monotonicity* (or its reverse) in the projection.

If the projection dimension is $p = 1$, one-dimensional embedding of numerical inputs forms a number line if the projections preserve monotonicity (resp. *reverse monotonicity*), i.e., for $x_1 < x_2$ (resp. $x_1 > x_2$), we have $f_{\text{LLM}}(x_1) < f_{\text{LLM}}(x_2)$. This ensures that the natural order of numerical values is maintained in the representation space.

To measure monotonicity properties of the function $f_{\text{LLM}}$ we use *Spearman rank correlation* that we briefly describe next. Let $X, Y \in \mathbb{R}^n$ be two real $n$-dimensional vectors and let $R(X)$ (resp. $R(Y)$) denote an $n$-dimensional vector obtained from $X$ (resp. $Y$) where the entries are substituted with their ranks in the sequence of sorted entries of $X$ (resp. $Y$). Then, Spearman rank correlation coefficients (usually denoted by $\rho$) is given by:

$$\rho = \frac{\mathbf{Cov}(R(X), R(Y))}{\sigma(R(x)) \cdot \sigma(R(Y))}, \tag{2}$$

where $\mathbf{Cov}(R(X), R(Y))$ is the covariance between rank vectors $R(X)$ and $R(Y)$, while $\sigma(R(X))$ and $\sigma(R(Y))$ are their respective standard deviations.

Spearman coefficient $\rho$ is a nonparametric measure for the alignment of the two vectors. Since projections can be flipped, we report the absolute value $\rho := |\rho|$ of the coefficient to assesses if the increment in one variable corresponds to the increase (or decrease) of the other.

### 3.3 PROPERTY 2: SCALING BEHAVIOR

To complement monotonicity, we define a novel measure of internal scaling in LLMs: the *Scaling Rate Index* (SRI), denoted by $\beta$. This index quantifies how the differences between numerical values are preserved or distorted under the model's transformation $f_{\text{LLM}}$.

The intuition is as follows: if $f_{\text{LLM}}$ preserves scale linearly, then adjacent representations of increasing numbers should be equally spaced. If instead the model compresses or expands numeric values (as hypothesized in cognitive theories of the mental number line), then these differences will vary in a structured way. In particular, we let $\beta$ be the parameter of geometric regression that fits the observed spacing between monotonic representations of given sequence $\{x_i\}_{i=1}^n$:

$$\min_{\alpha, \beta > 0} \sum_{i=1}^{n} \left| (y_{i+1} - y_i) - \alpha \beta^i \right|^2 \tag{3}$$

where $\{y_i\}_{i=1}^n := \{f_{\text{LLM}}(x_i)\}_{i=1}^n$. The motivation for this geometric form comes from the fact that convex and concave sequences (in the sense of second-order differences) can be naturally described using geometric progressions. For background on convexity and concavity, see Appendix A and Rockafellar (2015). The fitted value of $\beta^2$ characterizes the internal scaling of the representations, where the interpretation depends on the convexity/concavity of the sequence, as detailed in the experiments 4.1 and 4.2.

## 4 EXPERIMENTS

### 4.1 EXPERIMENT 1: IDENTIFYING NUMBER LINE USING CONTEXTUALIZED NUMBERS

The goal of this experiment is twofold. We first investigate whether LLMs encode numerical values along a *monotonic number line* (measured by $\rho$) in their internal representation space. Second, we test whether this proposed number line exhibits *sublinear scaling* (measured by $\beta$).

---

[2]We ignore the fitted $\alpha$ as it represents a global scaling factor and does not affect shape or compression properties.

To systematically probe the model's numerical representations, we partition numbers into logarithmically spaced groups:

$$G_1 = \{1, 2, \ldots, 20\},$$
$$G_{i+1} = \{10^i - 19, \ldots, 10^i + 20\}, \quad i \geq 2. \tag{4}$$

We use exponential groupings because they align with our hypothesis: if LLMs internally compress number space logarithmically, then equal steps in log-scale should produce structured patterns in hidden space. In contrast, uniform groupings which would over-represent small values and under-sample higher magnitudes. Further to ensure a representative sampling, we randomly select $k$ numbers from each group $G_i$, to avoid artifacts from local frequency or tokenization biases.

To analyze the embeddings of numbers, for every number $x \in G_i$, we assign the following prompt:

$$x \leftarrow \texttt{a=a, b=b, c=c, x=} \tag{5}$$

where $\texttt{a}$, $\texttt{b}$, and $\texttt{c}$ are randomly generated numbers from the groups $G_i$. This prompt structure is designed to provide the model with contextual examples, encouraging it to invoke the number $x$ in model's hidden states representations (see Figure 1). Such approaches have been used in prior work to probe contextual representations in language models Srivastava et al. (2024).

The transformation of hidden state $f_{\text{LLM}}(x)$ is extracted from a designated layer of the model from the last token in the prompt, i.e. the '=' token. We use only those $x$ for which the generated output of the model is $x$ itself. The set of hidden state representations, $\{f_{\text{LLM}}(x)\}_{x \in X}$, is then aggregated and analyzed to investigate patterns and properties in the embedding space. Proposition 1 interpret the meaning of the $\beta$ in this setting.

**Proposition 1.** *Let $f : \mathbb{R}_{\geq 0} \to \mathbb{R}$ be a strictly increasing function, and define the sequence $x_i := 10^i$ with $y_i = f(x_i)$ (here $f$ representing $f_{LLM}$). Suppose*

$$y_{i+1} - y_i = \alpha \cdot \beta^i$$

*for some constants $\alpha > 0$ and $\beta > 0$. Then:*

- *If $\beta = 1$, $f$ is a logarithmic function on $x_n$.*

- *If $\beta > 1$, $f$ is a super-logarithmic function on $x_n$.*

- *If $\beta < 1$, $f$ is a sub-logarithmic function on $x_n$.*

The intuition here comes clear as we calculate the limit

$$c := \lim_{i \to \infty} \frac{f(x_i)}{\log_{10} x_i}, \tag{6}$$

and notice that if $\beta = 1$, $c$ is a positive real constant, while if $\beta > 1$, $c$ is infinity, and finally, if $\beta < 1$, $c = 0$.

**Controlled Variant of Experiment 1** To control for potential biases introduced by tokenization where larger numbers often span more tokens, we conduct a control version on experiment 1. we conduct a complementary experiment using non-numerical sequences. Instead of numerical inputs, we construct sequences of random letters with lengths corresponding to the tokenized representations of numbers in equation 4. The letter sequences are grouped by their lengths so that the grouping approximately matches one of the numbers, and the prompts corresponding to specific letter sequences are designed in a similar fashion as for the numbers (5). This setting allows us to compare any observed structural patterns between the number representations and letter representations. By doing so, we can determine whether the model truly encodes numerical magnitude or if it is simply responding to surface-level features of the input.

### 4.2 EXPERIMENT 2: IDENTIFYING NUMBER LINE USING REAL-WORLD DATA

In the previous experiment (Section 4.1) we created an artificial experimental setting to test our hypothesis. In this experiment, however, we aim to further validate our hypothesis using real-world data. We collect names of celebrities along with their birth years and population of different cities/countries from Wikidata (Vrandečić & Krötzsch, 2014).

We prompt the model to provide the exact birth year or population size for each entity in a 1K-sample dataset, including notable individuals and countries (e.g., "What is the birth year of Ada Lovelace?" or "What is the population of Brazil?"), and filter the outputs to retain only valid numerical responses. From each valid prompt-response pair, we extract the hidden state at the position of the question mark across all layers. These hidden states are then used to train one- and two-component PLS models to regress on the gold numerical answers, and are also projected via PCA for dimensionality reduction.

For each layer and projection method, we compute $\rho$ and $\beta$, to quantify the internal scaling of numerical representations. We interpret the fitted value of $\beta$ according to the following remark, which characterizes the sublinear, linear, or superlinear behavior of geometric sequences.

*Remark* 1. In similar spirit to Proposition 1, let $f : \mathbb{R}_{\geq 0} \to \mathbb{R}$ be an increasing function with the sequence $x_i := i$ such that $y_i = f(x_i)$ (here $f$ representing $f_{LLM}$). Suppose

$$y_{i+1} - y_i = \alpha \cdot \beta^i$$

for some constants $\alpha > 0$ and $\beta > 0$. Then:

- If $\beta = 1$, $f$ is a linear function on $x_n$.

- If $\beta < 1$, $f$ is a sublinear function on $x_n$.

- If $\beta > 1$, $f$ is a strictly increasing function on $x_n$.

An immediate example of the linear case is $f(x_i) = x_i$, for which we can take $\alpha = 1$ and $\beta = 1$. This version of Proposition 1 will come useful when we perform experiments on real-world numerical data (see Section 5.2).

## 5 RESULTS

### 5.1 EXPERIMENT 1 RESULTS

The results reveal distinct yet consistent patterns in how different models encode numerical and alphabetical structures, with variations across layers (Table 1). Despite these variations, similar trends emerge across the models, leading to consistent conclusions about their processing of numerical values (please refer to Appendix B for experimental details).

First key finding is that numerical embeddings exhibit a significantly higher explained variance ($\sigma^2$ and $R^2$ in Table 1) in the one-dimensional PCA and PLS transformations compared to letter-based embeddings. This suggests that numbers naturally align along a one-dimensional manifold, akin to a number line, while random sequences of letters do not display the same structured behavior. However, to avoid artifacts of the architectural biases, an intervention study is completed along random directions and compared to these low-dimensional number lines in Appendix C.1 .

Furthermore, the monotonicity metric ($\rho$) consistently shows higher values for numerical data compared to the control experiment on alphabetical letters, with most models achieving $\rho > 0.9$ in both PCA and PLS analyses. This supports the idea that numerical representations are not only structured, but also maintain a well-ordered progression across layers. The resulting projections obtained using PCA for the numerical and letters groups are visualized in Figures 2a and 2b, respectively.

The sublinearity coefficient ($\beta$) derived from PCA projections reveals notable differences across models. Some, such as LLaMA-2-7B, Pythia, and GPT-2 Large, exhibit strong sublogarithmic (sublinear) scaling with $\beta < 1$, indicating that embedding distances grow at a diminishing rate. In contrast, models like Mistral show a nearly logarithmic trend ($\beta \approx 1$), while others approach a more linear spacing pattern with higher $\beta$ values.

In addition to Table 1, Figure 3 provides a layer-wise analysis for four models, demonstrating how sublinearity evolves across different depths. This analysis indicates that the explained variance is high at multiple layers. However, intervention along the PCA dimension in these layers reveals that not every layer is causally linked to the output (ref. Figure 6 in Appendix C.4). Finally, Figure 4 show the trends of metrics changing the number of context examples, showing how the models tends to converge with increasing number of examples in the context.

| Model | Group | PCA ($p = 1$) | | | | PLS ($p = 1$) | | | |
|---|---|---|---|---|---|---|---|---|---|
| | | Layer | $\rho \uparrow$ | $\beta$ Pr.1 | $\sigma^2 \uparrow$ | Layer | $\rho \uparrow$ | $\beta$ Pr.1 | $R^2 \uparrow$ |
| Llama-2-7B | Numbers | 3 | $0.97 \pm 0.00$ | $0.83 \pm 0.06$ | $0.60 \pm 0.01$ | 5 | $0.93 \pm 0.00$ | $2.62 \pm 0.00$ | $0.81 \pm 0.00$ |
| | Letters | 1 | $0.45 \pm 0.00$ | $1.21 \pm 0.00$ | $0.24 \pm 0.00$ | 27 | $0.88 \pm 0.03$ | $0.91 \pm 0.02$ | $0.45 \pm 0.01$ |
| Pythia-2.8B | Numbers | 8 | $0.94 \pm 0.01$ | $0.54 \pm 0.01$ | $0.31 \pm 0.01$ | 1 | $0.78 \pm 0.02$ | $4.65 \pm 1.32$ | $0.71 \pm 0.01$ |
| | Letters | 11 | $0.89 \pm 0.01$ | $0.53 \pm 0.10$ | $0.16 \pm 0.01$ | 20 | $0.90 \pm 0.01$ | $0.95 \pm 0.11$ | $0.46 \pm 0.04$ |
| GPT-2-L | Numbers | 18 | $0.95 \pm 0.00$ | $0.58 \pm 0.02$ | $0.32 \pm 0.00$ | 17 | $0.96 \pm 0.01$ | $1.15 \pm 0.09$ | $0.67 \pm 0.03$ |
| | Letters | 5 | $0.11 \pm 0.05$ | $0.80 \pm 0.42$ | $0.21 \pm 0.01$ | 33 | $0.81 \pm 0.03$ | $0.93 \pm 0.04$ | $0.44 \pm 0.01$ |
| Mistral-7B | Numbers | 3 | $0.96 \pm 0.00$ | $1.05 \pm 0.00$ | $0.44 \pm 0.00$ | 7 | $0.88 \pm 0.00$ | $14.87 \pm 8.28$ | $0.81 \pm 0.00$ |
| | Letters | 14 | $0.89 \pm 0.00$ | $0.60 \pm 0.00$ | $0.22 \pm 0.00$ | 29 | $0.86 \pm 0.00$ | $1.62 \pm 0.00$ | $0.63 \pm 0.00$ |
| Llama-3.1-8B | Numbers | 1 | $0.41 \pm 0.04$ | $1.14 \pm 0.05$ | $0.48 \pm 0.01$ | 4 | $0.93 \pm 0.01$ | $2.00 \pm 0.01$ | $0.73 \pm 0.01$ |
| | Letters | 1 | $0.56 \pm 0.00$ | $0.16 \pm 0.07$ | $0.19 \pm 0.01$ | 16 | $0.93 \pm 0.01$ | $0.88 \pm 0.06$ | $0.45 \pm 0.02$ |
| Llama-3.2-1B-Instruct | Numbers | 4 | $0.93 \pm 0.02$ | $1.33 \pm 0.12$ | $0.35 \pm 0.01$ | 6 | $0.91 \pm 0.00$ | $1.93 \pm 0.05$ | $0.68 \pm 0.01$ |
| | Letters | 1 | $0.57 \pm 0.06$ | $0.47 \pm 0.08$ | $0.17 \pm 0.00$ | 10 | $0.93 \pm 0.00$ | $0.97 \pm 0.03$ | $0.45 \pm 0.03$ |
| Deepseek-base-7B | Numbers | 2 | $0.65 \pm 0.015$ | $0.97 \pm 0.008$ | $0.83 \pm 0.090$ | 12 | $0.81 \pm 0.020$ | $0.26 \pm 0.793$ | $3.62 \pm 0.261$ |
| | Symbols | 1 | $0.23 \pm 0.014$ | $0.67 \pm 0.017$ | $3.01 \pm 2.469$ | 27 | $0.50 \pm 0.070$ | $0.29 \pm 0.862$ | $1.05 \pm 0.055$ |
| Qwen-1.5-7B | Numbers | 4 | $0.53 \pm 0.004$ | $0.97 \pm 0.007$ | $0.80 \pm 0.087$ | 18 | $0.80 \pm 0.020$ | $0.30 \pm 0.853$ | $2.35 \pm 0.404$ |
| | Letters | 2 | $0.47 \pm 0.028$ | $0.05 \pm 0.096$ | $0.89 \pm 0.178$ | 29 | $0.48 \pm 0.042$ | $0.90 \pm 0.006$ | $1.02 \pm 0.097$ |

Table 1: The monotonicity $\rho$ and the SRI $\beta$ for PCA and PLS on the same model/group pairs ($p$ refers to the projection dimension). We select the layers with the highest $\sigma^2$/ $R^2$ and for those layers we present the corresponding ($\rho$ and $\beta$). $\uparrow$ indicates higher is better. Note: $\beta = 1$ indicates logarithmic spacing. Please refer to Proposition 1.

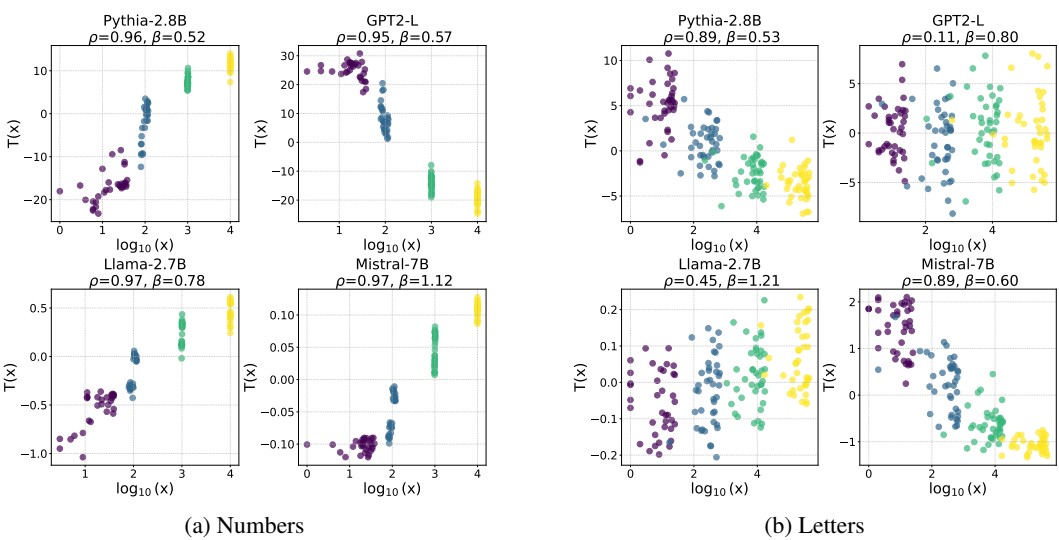

(a) Numbers          (b) Letters

Figure 2: Projections of hidden representations against log-scaled magnitudes for the layer with the highest explained variance in four models. (a) Numbers: consistent sublinear trends and strong monotonicity ($\rho$). (b) Letters: magnitudes proportional to length, also showing sublinearity and low monotonicity.

**PCA vs PLS.** The PLS method achieves high monotonicity ($\rho$) and explained variance ($R^2$) but exhibits lower sublinearity compared to PCA. This discrepancy arises because PLS operates as a supervised linear probe, where the regression target (e.g., numerical values) directly influences the projection. This process distorts the intrinsic spacing between points, as PLS prioritizes maximizing covariance with the target over preserving the original geometric structure. In contrast, PCA, being unsupervised, retains the relative spacing of data points in the latent space, better capturing the underlying sublinear trends. This distinction is evident in Table 1: PCA consistently reveals stronger sublinearity, while PLS achieves higher $R^2$ and $\rho$ by aligning the projection with the target variable.

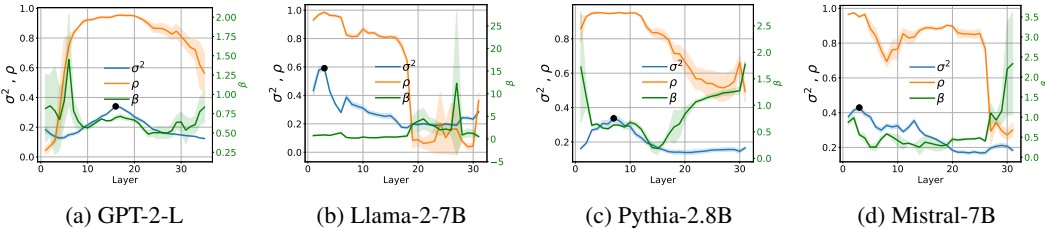

(a) GPT-2-L  (b) Llama-2-7B  (c) Pythia-2.8B  (d) Mistral-7B

Figure 3: Layer-wise analysis on numerical tokens across four models. Each panel shows explained variance ($\sigma^2$), monotonicity ($\rho$), and Scaling Rate Index ($\beta$) across layers. The maximal $\sigma^2$ typically aligns with a peak in $\rho$, highlighting optimal numerical encoding.

Notably, this aligns with findings in Zhu et al. (2025), where a linear probe failed to adequately capture the non-linear scaling of hidden states, particularly for larger numbers, where non-linearity becomes more pronounced. Our work explicitly quantify sublinearity using the Scaling Rate Index (SRI, $\beta$), which directly measures the rate of scaling in the latent space. This allows us to better capture the true geometric organization of numerical representations, especially in regimes where non-linear effects dominate.

## 5.2 EXPERIMENT 2 RESULTS

Building on the controlled, log-spaced setup from Experiment 1, we now test whether the same geometric regression objective applies to real-world numbers. We use two contrasting datasets: birth years (chronological, relatively clean) and country populations (noisy, context-dependent) and for our analysis, similarly as before we use Proposition 1 and its version in Remark 1.

For each model and dataset, we report the layer's average metrics ($\rho$ and $\beta$) under a 1D PCA/PLS probe (to control for decodability) and report the corresponding QA's accuracy (Table 2).

| Model | Group | Acc. | PCA ($p = 1$) | | | PLS ($p = 1$) | | |
|---|---|---|---|---|---|---|---|---|
| | | | $\overline{\rho} \uparrow$ | $\overline{\beta}$ Rm.1 | $\overline{\sigma^2} \uparrow$ | $\overline{\rho} \uparrow$ | $\overline{\beta}$ Rm.1 | $\overline{R^2} \uparrow$ |
| Mistral-7B | Birth years | 0.45 | 0.359 | 0.735 | 0.234 | 0.545 | 0.959 | 0.320 |
| | Populations | 0.000 | 0.112 | 1.672 | 0.868 | 0.221 | 1.060 | 0.027 |
| Deepseek-base-7B | Birth years | 0.490 | 0.259 | 0.849 | 0.200 | 0.655 | 0.949 | 0.290 |
| | Populations | 0.002 | 0.070 | 0.909 | 0.317 | 0.181 | 1.421 | 0.040 |
| Qwen-1.5-7B | Birth years | 0.325 | 0.124 | 0.998 | 0.509 | 0.587 | 1.064 | 0.312 |
| | Populations | 0.000 | 0.040 | 0.888 | 0.299 | 0.259 | 1.361 | 0.064 |
| Llama-2-7B | Birth years | 0.006 | 0.112 | 0.842 | 0.215 | 0.387 | 1.106 | 0.156 |
| | Populations | 0.000 | 0.195 | 1.118 | 0.147 | 0.314 | 1.231 | 0.033 |

Table 2: Natural-data number lines for birth years and populations. Accuracy is QA accuracy on each task. $\overline{\rho}$, $\overline{\beta}$, and $\overline{\sigma^2}$ / $\overline{R^2}$ are layer-averaged monotonicity, Scaling Rate Index, and explained variance for 1D PCA / PLS.

For the *birth year* task, most models exhibit strong trends, with relatively high monotonicity ($\rho$) and ($R^2$), while having low SRI ($\beta$), hence high compression. This indicates that the internal representations of birth years are well-structured and predictive, aligning with our expectations for numerical encoding in LLMs. On the other side, *Populations* shows low monotonicity score, hence the $\beta$ factor is not informative. We attribute the low monotonicity score to the non-structured internal representations of lower birth years, as can be seen in Figure 5.

For the *QA's accuracy*, birth years report higher accuracy by most models, suggesting birth years are answered more systematically. Unlike birth years, population figures are less context-dependent, influenced by geopolitical changes, reporting inconsistencies, and approximate expressions in text. Consequently, the low monotonicity makes the scaling ratio $\beta$ unreliable (details on experiment 2 can be found in appendix C.3).

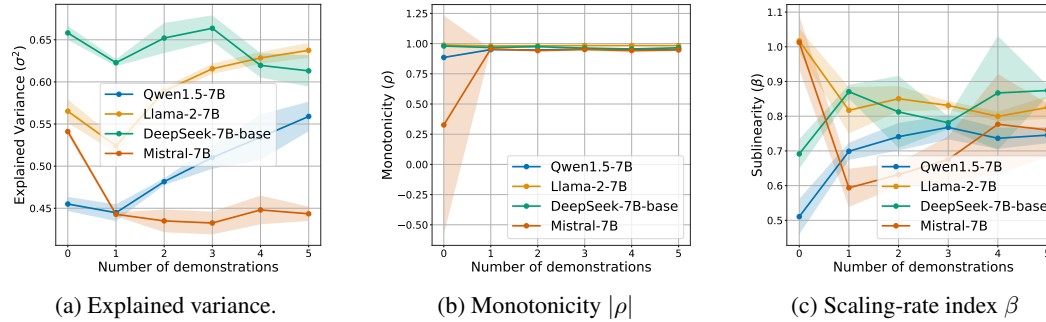

(a) Explained variance.  (b) Monotonicity $|\rho|$  (c) Scaling-rate index $\beta$

Figure 4: Generalization across prompt formats and # examples. Each panel sweeps the number of in-context examples $K \in \{0, 1, 2, 3, 4, 5\}$ for the equation-style ("$a=a$, $b=b$, ..., $x=$ ?") and plots mean±std% across three runs. (a) The explained variance of the PCA-derived 1D axis increases with $K$. (b) Order preservation $|\rho|$ strengthens with $K$. (c) The scaling-rate stabilizes in the sublinear regime ($\beta=1$ for log spacing).

Finally, Figure 5 illustrates representative one- and two-component PLS projections for two models on the birth-year dataset; in real-world numbers, these projections exhibit a pronounced nonlinearity in numeric spacing whereby years that are more frequently observed in training distributions (roughly 1800–2000) occupy disproportionately smaller arc-length on the learned "number line" (that is they are compressed) while earlier (rarer) years are expanded.

Intuitively, because PLS components are chosen to maximize covariance between hidden states and targets, variance concentrated in dense regions of the label space (recent centuries) is preferentially captured along the first components, effectively stretching distances among nearby, frequent labels and shrinking distances in sparse regions. The effect persists across both one- and two-component views, though it is more salient in the first component where most target-covariance is captured, and we observe the same qualitative pattern across deeper layers as well; see Appendix C.5 for comprehensive layer-wise projections and additional models. The first two images, concurrently with Table 2, visually show higher monotonicity index as can be seen in the smoother color transitions. The corresponding $\beta = 0.50$ shows the sublinear compression of the data.

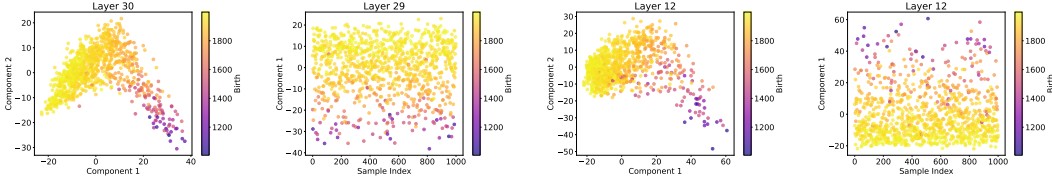

Figure 5: Visualization of PLS models trained on Llama-3.1-8B (first two images) and Llama-3.2-1B (last two images) model activations to predict entities' birth years using one and two dimensional PLS respectively. Each figure represents the layer with the highest $R^2$ score for one-component (second and fourth images) and two-component (first and third images) PLS models.

## 6 CONCLUSION

Inspired by the logarithmic compression in human numerical cognition, we investigate whether LLMs encode numerical values analogously. By analyzing hidden states across layers, we employ dimensionality reduction techniques (PCA and PLS) and geometric regression to test for two key properties: (1) order preservation and (2) sublinear compression, where distances between consecutive numbers decrease as values increase. Our results reveal that while both PCA and PLS identify numerical representations in a linear subspace, only PCA captures systematic sublinearity. This indicates that linear probes like PLS, which optimize for covariance with the target, may obscure the underlying non-uniform structure. Our findings suggest that LLMs encode numerical values with structured compression, akin to the human mental number line, but this is only detectable through methods like PCA that preserve geometric relationships.

ACKNOWLEDGMENTS

This work was written independently, with minor phrasing assistance from a large language model (ChatGPT).

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

APPENDIX

# A    CONCAVE AND CONVEX SEQUENCES

*Definition* 1. Let $(x_n)$, $n = 1, \ldots$ be a sequence of real numbers, and let us denote by $\Delta x_n := x_{n+1} - x_n$ and by $\Delta^2 x_n := \Delta x_{n+1} - \Delta x_n$.

- We say that $(x_n)$ is convex if $\Delta^2 x_n \geq 0$, for all $n$.

- We say that $(x_n)$ is concave if $\Delta^2 x_n \leq 0$, for all $n$.

- We say that $(x_n)$ is linear if $\Delta^2 x_n = 0$, for all $n$.

An example of a convex sequence that is relevant to our study is that defined with $x_i := 10^i$. Then, $x_{i+1} - x_i = 9 \cdot 10^i$. In the terminology of Section 3.3, we can take $\alpha = 9$ and $\beta = 10$.

An example of a concave sequence is given by $x_i = 1 - \frac{1}{10^i}$. In terminology of Section 3.3, we can take $\alpha = 9$ and $\beta = \frac{1}{10}$.

Finally, an example of a *linear* sequence is simply $x_i := i$, for which we can take $\alpha = 1 = \beta$.

# B    EXPERIMENTAL DETAILS

All experiments were performed using an NVIDIA A6000 GPU for accelerated computation. The models were implemented in Python and imported from Huggingface with PyTorch, and standard libraries like NumPy and Matplotlib were used for data processing and visualization. We evaluated the following models:

| Model | Variants | Ref. |
|---|---|---|
| Pythia | 2.8B | Touvron et al. (2023) |
| LLaMA | 2.7B, 3.1-8B, 3.2-1B | Touvron et al. (2023) |
| GPT-2 | Large-1.5B | Radford et al. (2019) |
| Mistral | 7B | Jiang (2024) |

Table 3: Models evaluated in the experiments.

Whenever possible, results were reported as the average of three runs, along with the standard deviation (std). For experiments where repeated runs were not feasible, the random seed was fixed to 42 to ensure reproducibility.

# C    ADDITIONAL EXPERIMENTS

## C.1    LAYERWISE CAUSAL INTERVENTION WITH PCA VS. RANDOM DIRECTIONS

To assess whether the number-line directions identified via PCA are causally active, we conducted controlled interventions on hidden states in three language models: LLaMA-2-7B, LLaMA-3.1-8B, and Mistral-7B-v0.1. For each model, we selected 120 numerical prompts of the form a=a, b=b, ..., x= and asked the model to generate the first token following the final equals sign.

At every transformer layer, we applied an additive shift along two types of directions: the PCA direction (PC1) and a randomly sampled direction from a spherical Gaussian. Each direction was normalized, and we swept across five shift magnitudes $\alpha \in \{-30, -15, 0, 15, 30\}$. The intervention took the form:

$$\mathbf{h}' = \mathbf{h} + \alpha \cdot \hat{\mathbf{v}}$$

where $\hat{\mathbf{v}}$ is the intervention direction. A trial was counted as successful if the output token following the = was a numeric string.

The results are summarized in Figure 6, where each subplot shows the number of successful generations (out of 120) as a function of the intervention layer. PCA-based interventions consistently

outperform random ones, particularly for negative $\alpha$ values. This indicates that the PCA direction captures a functionally meaningful dimension of number representation.

Interestingly, layers where PCA interventions were most effective tend to correspond to those with the highest explained variance in the PCA analysis (as previously shown in the monotonicity and sublinearity sections). This further supports the interpretation that the structure uncovered by PCA is not merely geometric but aligned with the model's internal abstraction of magnitude.

An asymmetry was observed: negative shifts (which move backward along the number line) produced stronger effects in earlier layers, whereas positive shifts tended to show more effect in later layers. This may reflect directional encoding of numerical scale or a compression effect where large-number abstraction is deferred to higher layers.

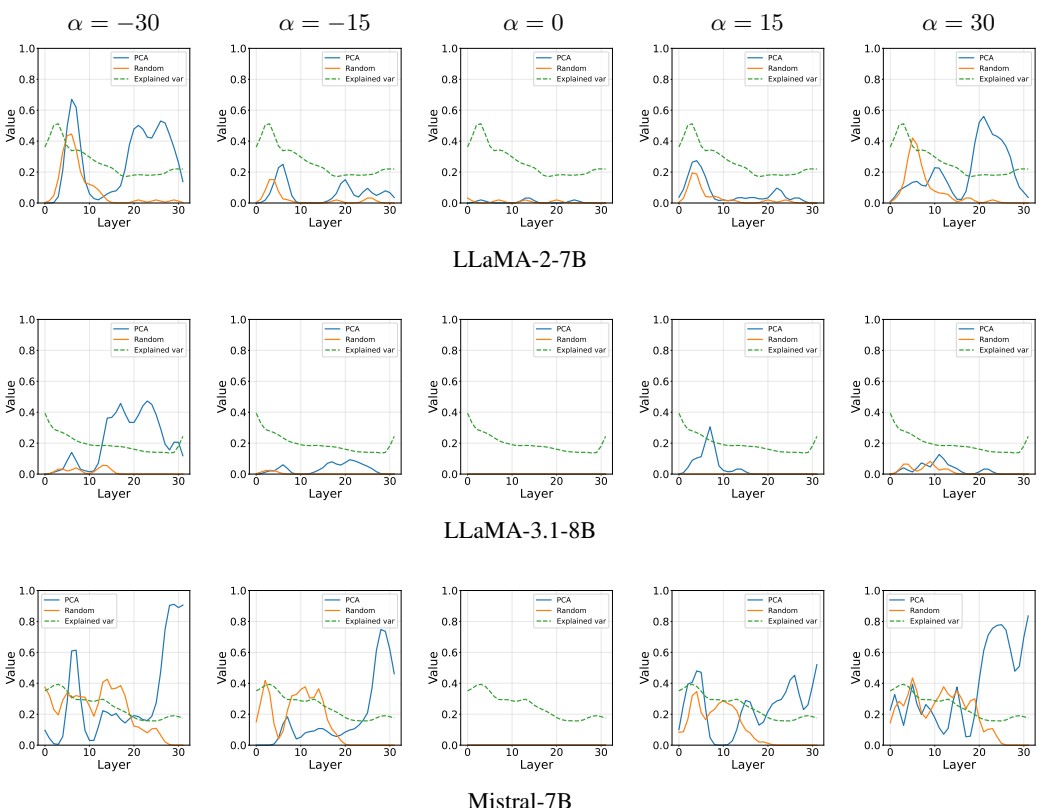

Figure 6: Causal intervention results for each model and shift magnitude $\alpha$. Each cell shows the number of numeric generations (out of 120 prompts) across transformer layers, comparing PCA vs. random direction interventions.

## C.2 Impact of sublinear magnitude coding on downstream numeric comparison

Our results indicate that numerals live on a *compressed* magnitude axis (Scaling-Rate $\beta < 1$) obtained by fitting a simple geometric regressor to PCA-projected hidden states. We now test whether this geometry has consequences for a basic downstream judgment: given two integers, *which is larger, **a** or **b**?* We draw pairs $(a, b)$ across scale bins defined by the midpoint $s = \log_{10}((a+b)/2)$; concretely, we use groups $10^i \pm 20$ so overall scale varies while the local gap $|a-b|$ can be held fixed (e.g., gaps in $\{1, 5, 10\}$). For each bin we sample 100 pairs and report both zero-shot and few-shot (3 very short demonstrations of the rule). Decoding is greedy ($T=0$). To avoid manual checking, we mark a response as correct if the first *content* token among the first 10 generated tokens is exactly a or b and matches the ground truth.[3]

Figure 7 shows four representative models in one row. Each subplot reports accuracy (y) versus scale bin (x), with zero-shot (dark blue) and few-shot (light blue) curves. A consistent picture emerges: when $|a-b|$ is fixed, accuracy often *declines with scale*. This is the signature of a compressed code, at larger magnitudes, equal numeric steps occupy less distance on the latent axis, shrinking decision margins. A small number of demonstrations largely flattens this slope and raises accuracy. The demonstrations act as local anchors that re-scale the relevant region without changing the underlying representation; where zero-shot and few-shot coincide at small magnitudes, the model already solves the task with sufficient margin.

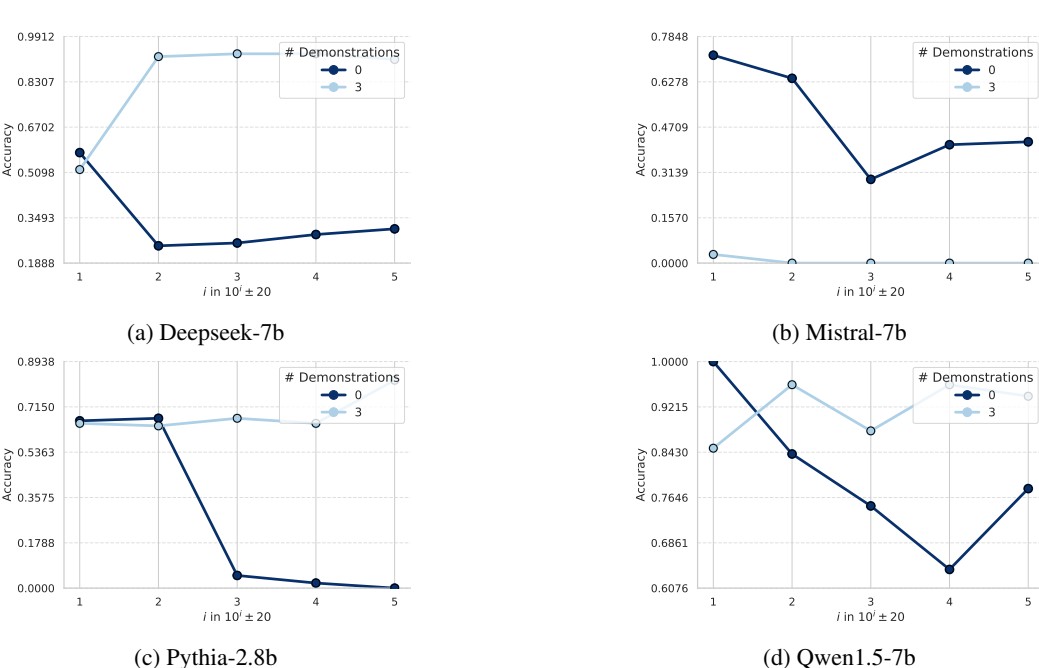

(a) Deepseek-7b

(b) Mistral-7b

(c) Pythia-2.8b

(d) Qwen1.5-7b

Figure 7: **Pairwise numeric comparison across four models.** Accuracy versus scale bin ($s = \log_{10}((a+b)/2)$) at fixed $|a-b|$. Dark: zero-shot. Light: few-shot ($k=3$). Compression predicts a downward zero-shot slope at larger scales; few-shot demonstrations act as anchors that increase effective separation and reduce scale sensitivity.

---

[3]Results are essentially unchanged if we instead score by the higher next-token logit between a and b, or if we strip punctuation/whitespace before reading the first content token.

## C.3 EXPERIMENT 2 PROTOCOL (BIRTH YEARS & POPULATIONS)

We test whether the sublinear geometry observed in controlled settings also appears in natural text. We use two factual QA slices: birth years and population sizes, and measure (i) QA accuracy and (ii) the geometry of the numeral representations. All prompts are zero-shot. We explicitly ask for digits only to avoid manual grading and to keep decoding deterministic.

*Prompt examples (zero-shot).*

```
What is the birth year of John, King of England?
Answer with digits only (no words, no punctuation).

What is the population size of Juab County?
Answer with digits only (no words, no punctuation).
```

We decode greedily ($T=0$) and read at most the first 20 new tokens. We locate the first token that consists only of digits; the predicted answer is the maximal contiguous digit span that follows. For representation analysis we take the hidden state of the token *immediately before* that first numeric token (our anchor into the context that triggers numeral production). If no digit token is generated in the first 10 steps, we fall back to the last input token. For QA, a year is correct if the 4-digit prediction matches the gold year exactly; population answers are compared after removing formatting (commas/spaces).

We group items by scale, accumulate predicted and gold answers to compute accuracy, and store per-layer vectors to derive a 1D axis (PCA as primary; PLS as supervised complement). On that axis we report $|\rho|$ (order) and $\beta$ (Scaling-Rate via adjacent gaps).

*Representative raw generations.*

```
Q: What is the birth year of John, King of England?
A: 1166 # correct (answer parsed from first digit span)

Q: What is the birth year of Isidore the Laborer?
A: The question is not clear. If you mean ... -> no early digits; fallback to last input
    token for states

Q: What is the population size of Juab County?
A: 10342 # correct

Q: What is the population size of Grunewald?
A: The population size is 100000 ... # parser extracts '100000'
```

This protocol keeps the natural-text slice deliberately simple: zero-shot prompts, deterministic decoding, automatic numeric parsing, and a clear rule for where we read a representation (the token just before the first numeric token).

## C.4 LAYER-WISE PLS ANALYSIS

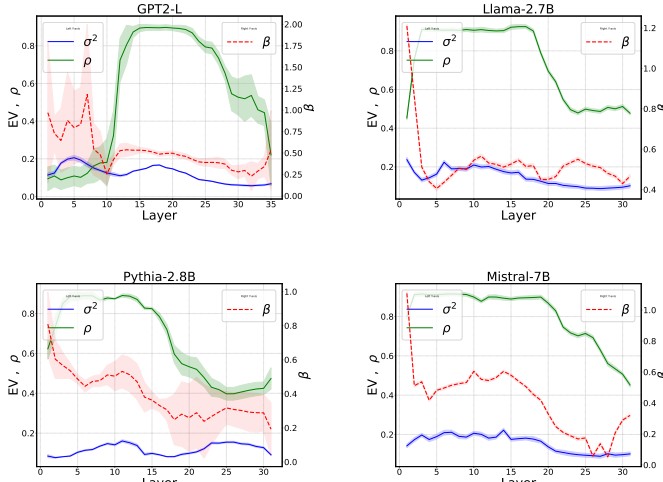

Figure 8: Layer-wise analysis of four models on letters groups, showing explained variance (EV or $\sigma^2$), monotonicity ($\rho$), and Scaling Rate Index ($\beta$).

## C.5 BIRTH YEAR AND POPULATION DATASETS PROJECTIONS IN ALL LAYERS

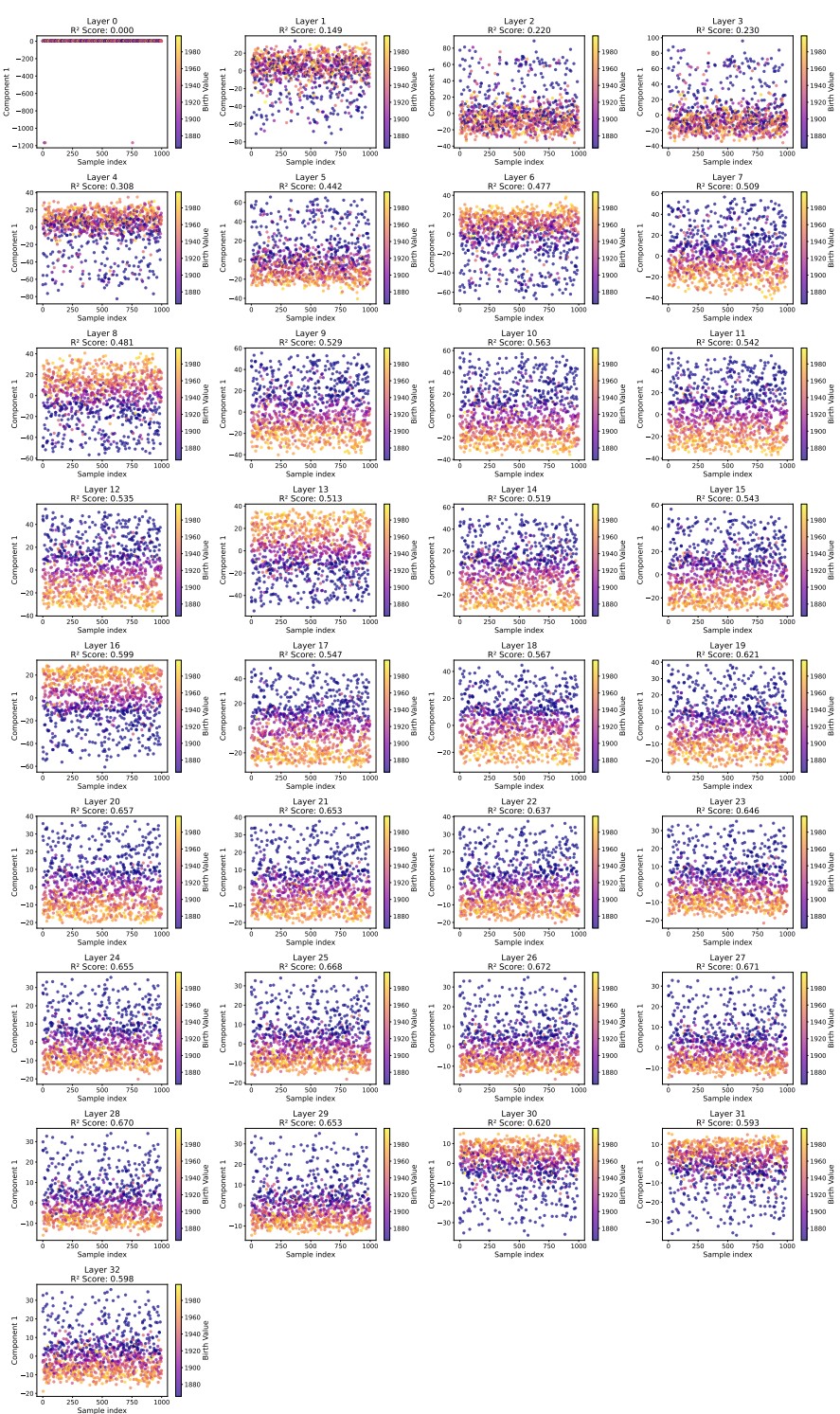

Figure 9: One component PLS model trained on Llama-3.1-8B instruct model activations to predict entities' birth year.

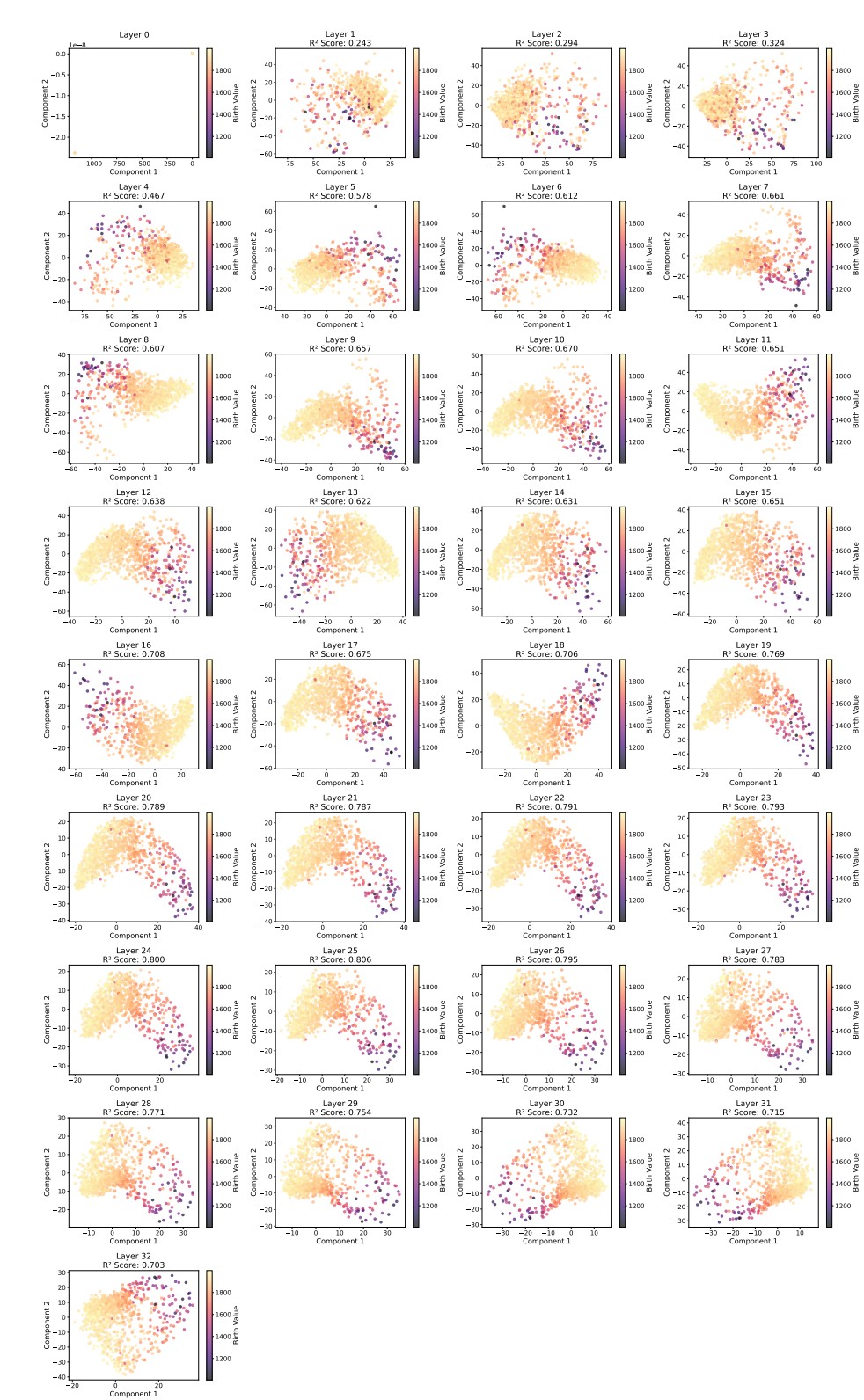

Figure 10: Two components PLS model trained on Llama-3.1-8B instruct model activations to predict entities' birth year.

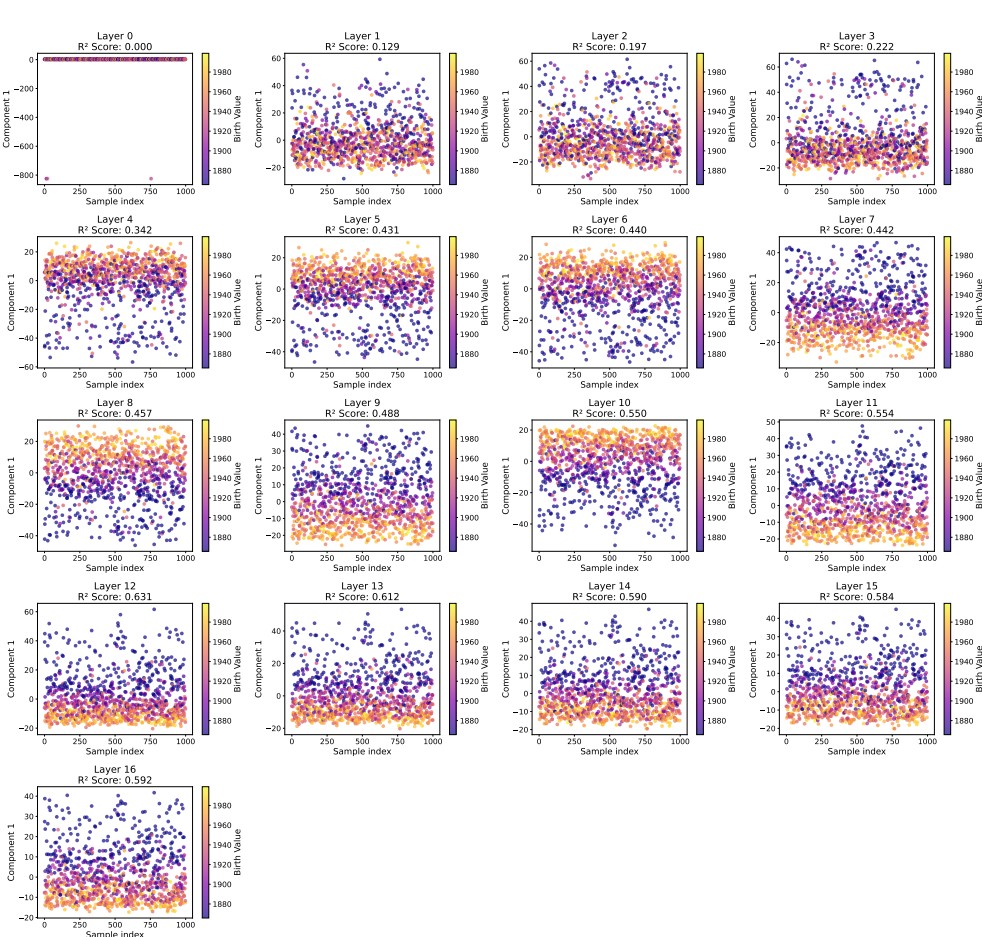

Figure 11: One component PLS model trained on Llama-3.2-1B instruct model activations to predict entities' birth year.

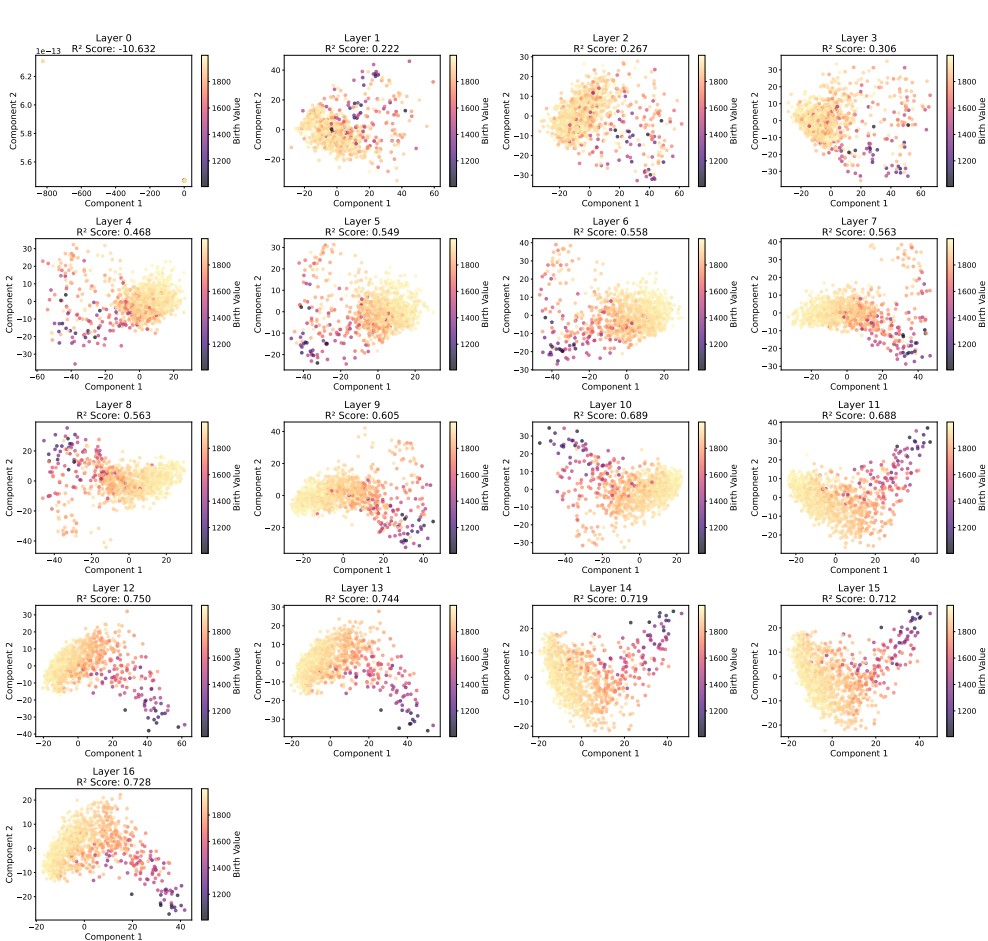

Figure 12: Two components PLS model trained on Llama-3.2-1B instruct model activations to predict entities' birth year.

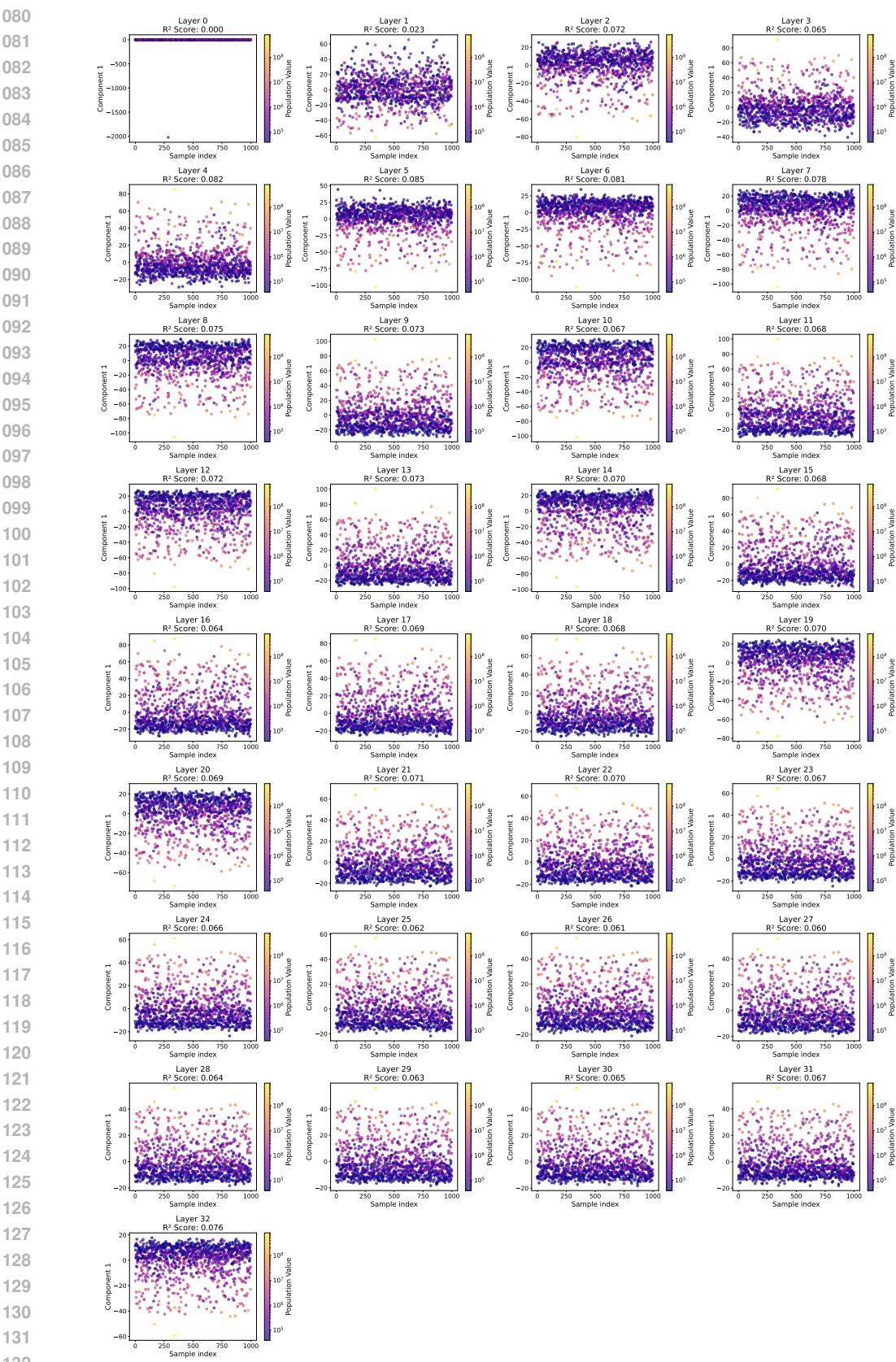

Figure 13: One component PLS model trained on Llama-3.1-8B instruct model activations to predict entities' population size.

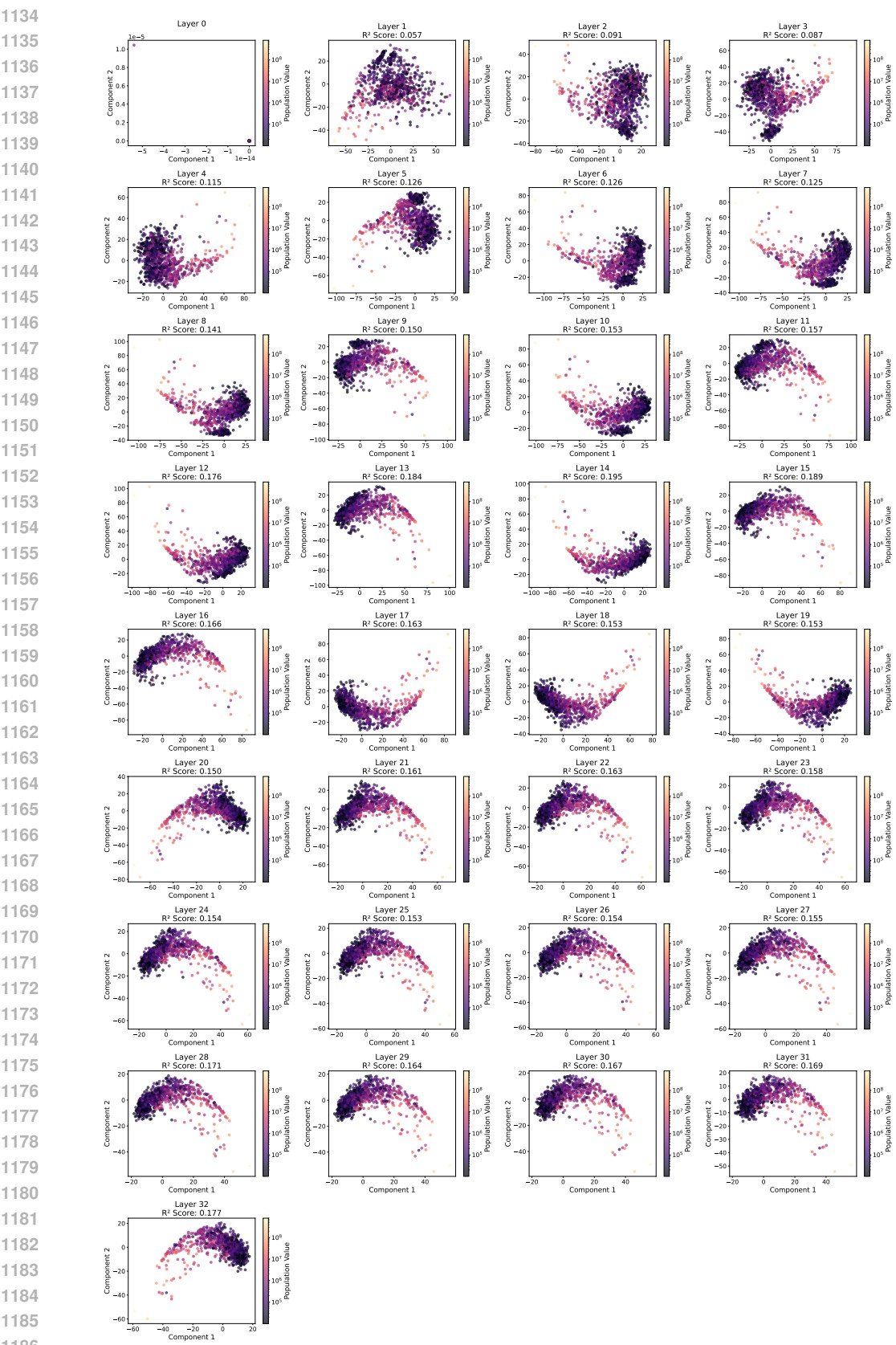

Figure 14: Two components PLS model trained on Llama-3.1-8B instruct model activations to predict entities' population size.

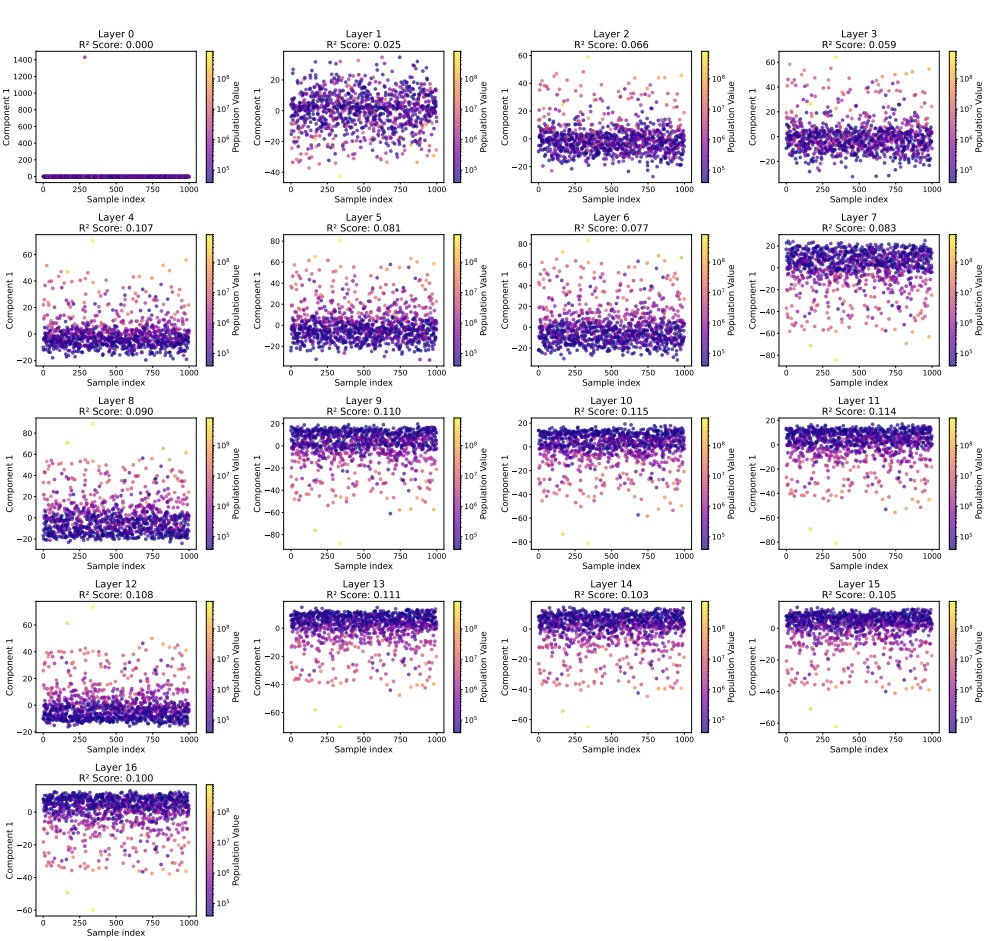

Figure 15: One component PLS model trained on Llama-3.2-1B instruct model activations to predict entities' population size.

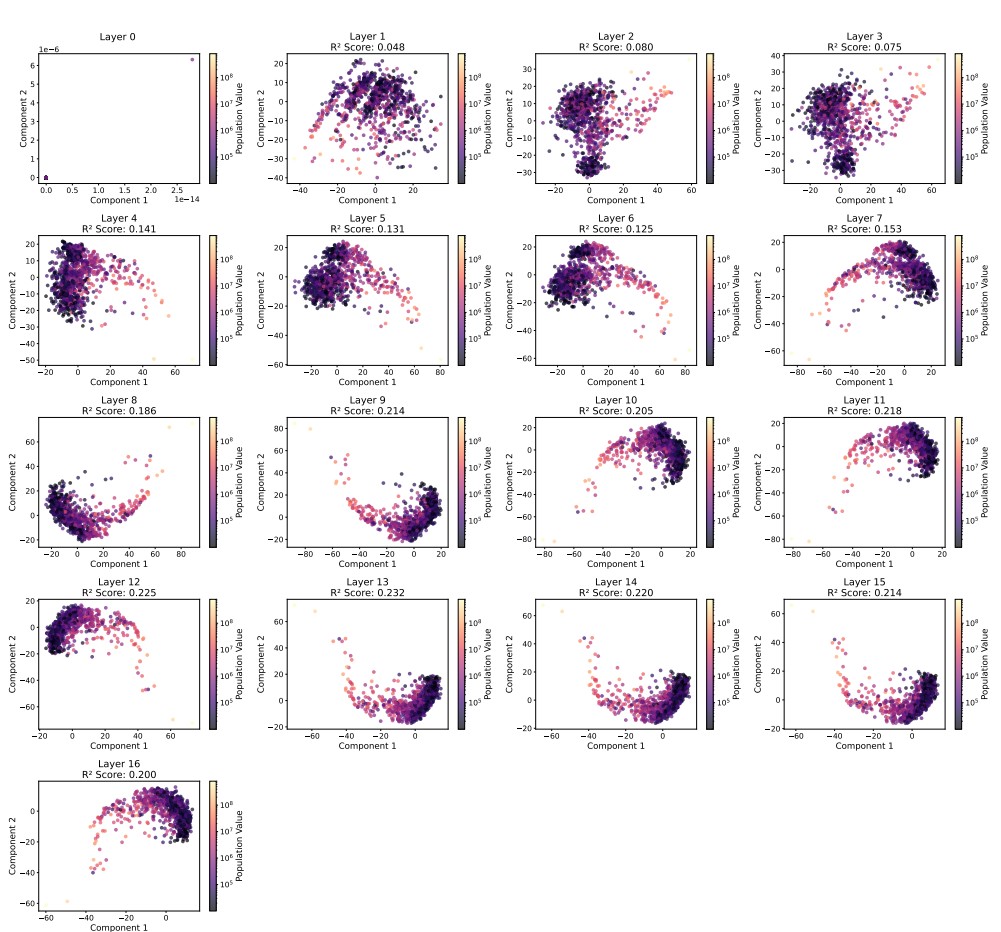

Figure 16: Two component PLS model trained on Llama-3.2-1B instruct model activations to predict entities' population size.

