# OpenReview forum: "NUMBER REPRESENTATIONS IN LLMS: A COMPUTATIONAL PARALLEL TO HUMAN PERCEPTION"
_ICLR.cc/2026/Conference — Submitted to ICLR 2026_

### Official Review · Reviewer_jbmZ · 2025-10-30

**Soundness:** 3
**Presentation:** 4
**Contribution:** 2
**Rating:** 4
**Confidence:** 3

**Summary:**

This paper considers whether LLMs encode numbers using logarithmic rather than linear representations. Specifically, the authors pair both a synthetic dataset and some simple real-world data with PCA and PLS to test for order preservation and sublinear compression of numeric representations. Their interpretation of their results is that LLMs do encode numbers using logarithmic representations, in a way that is consistenct with the human mental number line.

**Strengths:**

The question is interesting and well motivated. The writing is clearly structured and easy to follow. Experiments are simple but clean. Finding parallels between human numeric cognition and LLM numeric cognition can be illuminating into both systems.

**Weaknesses:**

My main concern with this paper is the lack of impact. Even though the topic and questions are interesting to me, I do not believe that the broader ICLR audience will find it impactful.

- The results on the natural data seem fairly weak and the use of birth years to study something analogous to mental number lines is not very compelling. Although birth years are numbers, they do not necessarily symbolize quantity in the way that most other numbers do. For example, '1984' in the input 'Katy Perry was born in 1984' is perhaps more similar to 'California' in the input 'Katy Perry was born in California' than it is to '1984' in the input '1700 + 284 = 1984'. Put another way, the numeric interpretation of '1984' seems far more meaningful in the latter case than the former case. The way that the birth years are presented the real data task therefore do not seem to lend themselves quite as naturally to some notion of having an internal number line.
- Building on the previous point, it is not clear why the real data experiments use factual questions that only have one correct numeric answer. This also seems to be a fairly unnatural context in which to consider mental number lines, since the model can just memorize that single number without needing to resort to any kind of numeric representation.

**Questions:**

- It is not clear how I should interpret specific values of ρ and β beyond some vague notion of 'higher is better'. Is there a concrete baseline that these values can be benchmarked against?
- In table 2, is 'llama' being misspelled as 'llamba'?

---

> ### Author Response · Authors · 2025-11-21
>
> > **Reviewer jbmZ**: My main concern with this paper is the lack of impact. Even though the topic and questions are interesting to me, I do not believe that the broader ICLR audience will find it impactful.
>
> We think of having logarithmic encoding (e.g. $\beta \leq 1$ in Exp.1) of numerical information as a proxy of more deeper insights on how these next-word-trained models tend to align with both the quantity and the nature of information.
>
>  **First**, under matched pipelines and prompts, non-numeric controls (e.g., letters) do not exhibit comparable compression, indicating the effect is not a by-product of tokenization or repetition but tracks magnitude semantics (Table 1; Fig. 2).  The deeper insight we like to share with the community is this supports a broader view that next-word training on large corpora tends to “compress the compressible,” trading global range for local resolution. **Second**, our geometry helps explain prior observations that linear probes deteriorate for large numbers [1] (cf. L43; detailed at L396): if equal numeric steps occupy progressively smaller distances on the learned axis, linear decoders face diminishing margins at higher scales.
>
> **Third**, to wrap up this useful discussion: we add a simple behavioral experiment where the model answers “which is larger, a or b?” . We sample pairs across scale bins defined by $10^i±20$ (same bins used in experiment 1), so scale varies while holding the gap $∣a−b∣$ fixed. We evaluate zero-shot and few-shot variants (see table below). Two patterns emerge: (i) at fixed $∣a−b∣$ , zero-shot accuracy decreases with scale, consistent with compression; and (ii) few-shot flattens this slope, acting as local anchors that increase effective separation without altering the underlying representation, but more importantly zero-shot and few-shot coincide at small magnitudes. Collectively, these results illustrate how the finding of descriptive geometry might be a useful diagnostic for scale-dependent reliability (refer to Appendix C.2 for details and visualization).
>
> | Model       |  Shots  |  G1  |  G2  |  G3  |  G4  |  G5  |
> | ----------- | :-----: | :--: | :--: | :--: | :--: | :--: |
> | Qwen1.5-7B  |  0-shot | 1.00 | 0.84 | 0.75 | 0.64 | 0.78 |
> | Qwen1.5-7B  | 3-shots | 0.85 | 0.96 | 0.88 | 0.96 | 0.94 |
> | pythia-2.8b |  0-shot | 0.66 | 0.67 | 0.05 | 0.02 | 0.00 |
> | pythia-2.8b | 3-shots | 0.65 | 0.64 | 0.67 | 0.65 | 0.82 |
>
> > **Reviewer jbmZ W1+2**: The results on the natural data seem fairly weak and the use of birth years to study something analogous to mental number lines is not very compelling..Although birth years are numbers, they do not necessarily symbolize quantity...
>
> We intentionally kept Experiment 2 small and descriptive because we partially agree with the reviewer. But it’s important to give a flavor of real-world text, not to serve as the core evidence. Our main insights (sublinear spacing and monotonicity) come from Experiment 1, which isolates numeral geometry under controlled prompts.
> Further, we chose correct birth years because they are ubiquitous, strictly ordered in time, and avoid unit/format confounds. We do not claim they behave like counts but only that, in everyday text, they provide an ordered magnitude. (see [2] for similar insights)
>
> > **Reviewer jbmZ Q1**: It is not clear how I should interpret specific values of ρ and β beyond some vague notion of 'higher is better'. Is there a concrete baseline that these values can be benchmarked against?
>
>  We interpret $\rho$ as Spearman’s rank correlation that quantifies order preservation (monotonicity) of the learned 1D embedding; because the axis can be flipped, we report $|\rho|\in[0,1]$. The $\beta$ coefficient is subtler: it summarizes the curvature of spacing via a geometric fit to adjacent gaps. As formalized in Proposition 1 and Remark 1, $\beta$ must be read relative to the input sequence. For linear sequences $x=1,2,3,\ldots$, $\beta=1$ corresponds to linear spacing in $y$; $\beta<1$ means the embedding is sublinear/concave (compressed), and $\beta>1$ means superlinear/convex (expanded), with $|1-\beta|$ indicating how strong the deviation is. For log-spaced sequences $x=10^i$, $\beta=1$ corresponds to logarithmic spacing; here $\beta<1$ indicates stronger-than-log compression, while $\beta>1$ indicates weaker-than-log spacing. We are not aware of a community benchmark scale that canonically maps specific $\rho,\beta$ values to grades.
>
> Thank you for your time and for helping us sharpen the contribution, and we are happy for reconsideration of rating or raising another question.
>
> [1] Zhu, F., Dai, D., & Sui, Z. (2025, January). Language models encode the value of numbers linearly. In the 31st International Conference on Computational Linguistics .
>
> [2] Heinzerling, B., & Inui, K. (2024, August). Monotonic Representation of Numeric Attributes in Language Models. In the 62nd Annual Meeting of the Association for Computational Linguistics (Volume 2: Short Papers).

---

> > ### Comment · Reviewer_jbmZ · 2025-11-25
> > **Raised score to a 6 in response to rebuttal**
> >
> > Upon greater reflection in response to the rebuttal, the non-linear relationship between compressed representations and a linear number system is profound. I've raised my score to a 6. Good luck to you!

---

> > > ### Author Response · Authors · 2025-11-27
> > >
> > > We thank the reviewer for their follow-up, and the positive reconsideration.

---

### Official Review · Reviewer_9VTM · 2025-11-01

**Soundness:** 3
**Presentation:** 3
**Contribution:** 3
**Rating:** 8
**Confidence:** 4

**Summary:**

This manuscript investigates how LLMs internally represent numerical values. The authors find that LLMs encode numbers on a compressed, logarithmic number line, rather than a linear one as commonly assumed. By extracting hidden states corresponding to numerals and projecting them onto one-dimensional manifolds, the study evaluates two metrics: Spearman’s ρ to assess monotonicity and a new Scaling Rate Index (β) to characterize the spacing pattern as sublinear, linear, or superlinear.

**Strengths:**

1. The target question is interesting. If number-related questions in the area of LLMs can be thoroughly understood, it may lead to further significant advancements.

2. The proposed methodology is well-motivated and interpretable.

3. The conclusion is highly inspiring, as it establishes a connection between cognitive psychology and representational analysis in LLMs.

**Weaknesses:**

1. The paper shows that compression exists, but not why (e.g., is it due to token frequency, positional embeddings, or training distribution?).A deeper analysis of architectural causes or training data statistics would strengthen the work.

2. It’s unclear how logarithmic compression affects numerical reasoning benchmarks or real-world performance (e.g., arithmetic, scale extrapolation).


3. Although multiple runs and controls are used, confidence intervals are sometimes narrow, and cross-model consistency could be better quantified.

**Questions:**

1. What is the causal origin of the logarithmic compression? Could it be attributed to the frequency distribution of numbers in the training corpora, which approximately follows a power-law?

2. Have you compared the compression patterns across different tokenization schemes (e.g., digit-based vs. word-based numerals) to rule out tokenizer-level artifacts?

3. Have the authors examined whether a similar logarithmic compression of numerical magnitude also emerges in MLLMs that jointly encode visual and textual inputs? For instance, do vision-language models exhibit comparable sublinear spacing when representing quantities inferred from visual stimuli? Investigating this direction could help determine whether logarithmic number representation is a general emergent property of large-scale multimodal learning, rather than a phenomenon restricted to linguistic models.

**Details Of Ethics Concerns:**

no.

---

> ### Author Response · Authors · 2025-11-21
>
> > **Reviewer 9VTM W1**: The paper shows that compression exists, but not why (e.g., is it due to token frequency, positional embeddings, or training distribution?)
>
> We agree this is an important question. However, finding the cause (token frequency, positional effects, training distribution...) is a substantial study on its own. Our paper’s scope is narrower but sufficiently studied:: to characterize the geometry (via $\rho$ and $\beta$) and show it has behavioral consequences (pairwise comparison). We will make this scope explicit and add a short “ Limitations” paragraph.
>
> > **Reviewer 9VTM W2**: It’s unclear how logarithmic compression affects numerical reasoning benchmarks or real-world performance (e.g., arithmetic, scale extrapolation).
>
> We added small, targeted checks tying geometry to behavior: performance of model in answering the questions "which is larger a or b?$ at fixed ∣a−b∣, we show that accuracy decrease with scale; few-shot flattens the slope (Refer to appendix C.2 for more details).
>
> > **Reviewer 9VTM W3**: Although multiple runs and controls are used, confidence intervals are sometimes narrow, and cross-model consistency could be better quantified.
>
> We will revise the paper with new models as suggested by reviewer “W9st” and we hope that enhances this point.
>
> > **Reviewer 9VTM Q2**Have you compared the compression patterns across different tokenization schemes (e.g., digit-based vs. word-based numerals) to rule out tokenizer-level artifacts?
>
>
> Within our scope, we provide two pieces of evidence that the effect is not a tokenizer-level artifact:
> Non-numeric controls (letters). Using length token-count–matched letter sequences under the same pipeline, we see no compression, whereas numerals show $\beta <1$. This points to a quantity-specific effect rather than tokenization or repetition (Table 1; Fig. 2).
>
>
> Cross-model replication (different vocabularies). We repeat the geometry on multiple model families with distinct vocabularies/tokenization choices (e.g., Pythia-2.8B, Mistral-7B, Qwen-1.5-7B, DeepSeek-7B-base). The sign of curvature is stable across most models, with similar magnitudes under the same prompt/processing setup.
>
> > **Reviewer 9VTM Q3** Have the authors examined whether a similar logarithmic compression of numerical magnitude also emerges in MLLMs that jointly encode visual and textual inputs?
>
> We have not evaluated MLLMs in this submission. But we agree with reviewer it is very important to study these models in future work.
>
> Thank you for the positive assessment and for the sharp questions, and we are happy for any further questions.

---

### Official Review · Reviewer_W9st · 2025-11-01

**Soundness:** 2
**Presentation:** 3
**Contribution:** 2
**Rating:** 4
**Confidence:** 3

**Summary:**

This paper challenges the prevailing linear representation hypothesis, arguing instead that LLMs encode numerical values on a compressed, logarithmic number line. To test this, the authors extract hidden states for numerals, project them using dimensionality reduction, and then measure monotonicity and spacing using a novel Scaling Rate Index. Their analysis consistently reveals strong sublinear trends across various LLM families, indicating a compressed, non-uniform geometry. The authors validate this finding through causal interventions that modulate next-number predictions and demonstrate its robustness in real-world settings while showing its absence in non-numerical controls.

**Strengths:**

1. The authors tackled the novel and underexplored problem of how LLMs internally represent and manipulate numeric values.
2. The relevant papers and theories are well-introduced in the manuscript, which helps the reader to easily follow the logic of the proposed method.
3. Two properties, monotonicity and scaling, are well defined and analyzed with appealing metrics.

**Weaknesses:**

1. The authors experimented with a single prompt, which is very unlikely to appear in a real-world scenario. Therefore, it is unclear how the experimental findings in this paper can be generalized to real-world data. To improve generalizability, the authors should consider using simple prompts with single numbers (e.g., "2047"), varying the number of preceding in-context examples (e.g., i) "2047=2047 104=?", ii) "2047=2047 104=104 37=?"), or experimenting with numbers from grade-school math QA pairs such as GSM8K, ASDiv, or other datasets. By conducting the same analysis on diverse prompts and reporting the results with statistical analysis, the authors' contribution would be more concrete.

2. The models used in the paper are limited in their numerical reasoning capabilities. Even if the authors are not able to utilize commercial, closed-source models, open-source and light-weight reasoning models such as Qwen, DeepSeek, GPT-OSS, and any other model families that show better results on numerical reasoning are available. I believe that including models with stronger numerical reasoning abilities would help reveal whether the poor understanding of numbers is a general limitation of LLMs. Moreover, including bidirectional language models (e.g., diffusion LMs) might be beneficial. Because the number is always placed at the end of the prompt in the probe's current design, this could introduce positional bias. In bidirectional LMs, the position of the number can be varied, which would be free from such unintended bias.

3. Since PCA and PLS convey different analysis results, I wonder how robust the analysis results proposed in the paper are. For example, if another dimensionality reduction method were utilized, would the analysis results change? If so, how can we say that PCA and PLS are the most effective dimensionality reduction algorithms for analyzing number representations?

**Questions:**

1. In Table 1, why do the authors compare results from different layers? For LLaMA-3.1-8B, comparing the monotonicity metric and sublinearity coefficient from the same layer seems fairer. If this is not the case, it would be better if the authors elaborated on the reason for this in the manuscript.

2. Why were only LLaMA models used for experiment 2?

---

> ### Author Response · Authors · 2025-11-21
>
> > **Reviewer W9st W1**: The authors experimented with a single prompt, which is very unlikely to appear in a real-world scenario. Therefore, it is unclear how the experimental find.. To improve generalizability, the authors should consider using simple prompts with single numbers (e.g., "2047"), varying the number of preceding in-context examples (e.g., i) "2047=2047 104=?", ii) "2047=2047 104=104 37=?"),...
>
> To clarify, our paper already evaluates the equation-style template (“a=a, b=b, …, x=”) as the reviewer suggests, and ablate on the number of in-context examples (we report trends for K=1–4 in Figure 4), however following reviewer suggestions we include k=0, which shows that it's not easy to make the model think of numerical values without any example (please refer to figure 4).
> In addition, we run the same analysis on natural QA-style prompts drawn from real data (e.g., birth years, populations), again observing consistent structure for birth years.
> Regarding GSM8K, even ostensibly number-centric questions frequently introduce units and scales—e.g., ‘37 dollars’, or ‘in 5 years’. These cues encourage the model to recruit unit-bound priors and operations (addition with time, conversion), which can blur the signal we attribute to pure numeral geometry.
>
> > **Reviewer W9st W2**:The models used in the paper are limited in their numerical reasoning capabilities...
>
> In the revision we broaden coverage beyond our original set by adding two strong, lightweight open models : Qwen1.5-7B and DeepSeek-7B, and report both the geometry ($\beta$,$\rho$) and the downstream pairwise comparison probe where the main patterns persist (please refer to table 1, and Appendix C2 for details) . For the bidirectional models, our setting can't translate there, as we discussed in first experiments, we make the model think of the number x by casting it in the prompt a=a,b=b,c=c,x=. However, while this is possible in auto-regressive models, its not in non-auto-regressive models.
>
> | Model                | Set      | Layer | EV (±)    | rho (±)    | beta (±)     |
> | -------------------- | -------- | :---: | ------------ | ------------ | ------------ |
> | deepseek-llm-7b| numerics |   2   | 0.65 ± 0.015 | 0.97 ± 0.008 | 0.83 ± 0.090 |
> | deepseek-llm-7b| letters  |   1   | 0.23 ± 0.014 | 0.67 ± 0.017 | 3.01 ± 2.469 |
> | Qwen1.5-7B           | numerics |   4   | 0.53 ± 0.004 | 0.97 ± 0.007 | 0.80 ± 0.087 |
> | Qwen1.5-7B           | letters  |   2   | 0.47 ± 0.028 | 0.05 ± 0.096 | 0.89 ± 0.178 |
>
>
>
>
> > **Reviewer W9st W3**: Since PCA and PLS convey different analysis results, I wonder how robust the analysis results proposed in the paper are.
>
> Thank you for raising this. Our view is that the right dimensionality reduction depends on the purpose, what we needed is a 1D axis that reflects the intrinsic geometry of numeral representations so we can measure order ($|\rho|$) and spacing curvature ($\beta$).
> PCA is unsupervised and variance-preserving, so it reveals the geometry of hidden states without being pulled toward a target. This makes it appropriate for spacing analysis (our main goal). On the other hand, PLS is supervised (aligns to a target like $x$ or $\log x$); it’s useful to expose a task-aligned axis (e.g., for causal steering), but it can distort spacing because it explicitly optimizes covariance with labels. We therefore report PLS as a complement, not as the primary geometry.
> Across models and prompts, we find the sign of curvature is stable: $\beta<1$ (sublinear) in $>95%$ of runs across methods; $|\rho|$ remains high.
> Finally, We do not claim PCA/PLS are “the most effective” in a universal sense; rather, PCA is the appropriate primary tool for intrinsic spacing, while PLS is appropriate when a supervised, controllable axis is desired.
>
> > **Reviewer W9st Q1**:In Table 1, why do the authors compare results from different layers? For LLaMA-3.1-8B, comparing the monotonicity metric and sublinearity coefficient from the same layer seems fairer. If this is not the case, it would be better if the authors elaborated on the reason for this in the manuscript.
>
>  It's important to note that, for PCA we select the layer with max explained variance; for PLS the layer with max $R^2$, this is why each model in table 1 is reported different layers. The logic behind this is that if we are reporting numerical properties (monotonicity and sublinearity, the information should be first laying in 1-D with enough support). Further intervention analysis is is provided in appendix C.1 discussing how layers with high explained variance are causally related to the output numeric values.
>
> > **Reviewer W9st Q2**: Why were only LLaMA models used for experiment 2?
>
> Thanks for flagging this. Experiment 2 was intentionally lightweight and our initial natural-text pipeline was built around the LLaMA family, so we reported those first.
>
> We thank the reviewer for their time, and we would be grateful if these changes resolve your concerns and update the rating.

---

> ### Author Response · Authors · 2025-12-01
>
> > **Update on Table 2 (broader model coverage regarding reviewer's Q2).**
>
>  In the revision we expanded Table 2 to include Mistral-7B, DeepSeek-7B, and Qwen-1.5-7B alongside Llama-2-7B. The main patterns replicate across families: for birth years, PCA shows meaningful*sublinear spacing (β<1 on average), while populations remain less ordered (consistent with the QA's accuracy and model-strength). We report layer-averaged metrics ($\rho, \beta, \sigma^2,R^2$) and the QA accuracies. This addresses the concern that our natural-data results were LLaMA-specific.
>
> **Natural-data number lines (birth years & populations).**
> Accuracy = QA accuracy. Metrics are layer-averaged: ρ (monotonicity), β (Scaling Rate Index), σ² (PCA explained variance), R² (PLS).
>
> | Model            | Group       |  Acc. | PCA ρ | PCA β | PCA σ² | PLS ρ | PLS β | PLS R² |
> | ---------------- | ----------- | :---: | :---: | :---: | :----: | :---: | :---: | :----: |
> | Mistral-7B       | Birth years | 0.450 | 0.359 | 0.735 |  0.234 | 0.545 | 0.959 |  0.320 |
> | Mistral-7B       | Populations | 0.000 | 0.112 | 1.672 |  0.868 | 0.221 | 1.060 |  0.027 |
> | Deepseek-base-7B | Birth years | 0.490 | 0.259 | 0.849 |  0.200 | 0.655 | 0.949 |  0.290 |
> | Deepseek-base-7B | Populations | 0.002 | 0.070 | 0.909 |  0.317 | 0.181 | 1.421 |  0.040 |
> | Qwen-1.5-7B      | Birth years | 0.325 | 0.124 | 0.998 |  0.509 | 0.587 | 1.064 |  0.312 |
> | Qwen-1.5-7B      | Populations | 0.000 | 0.040 | 0.888 |  0.299 | 0.259 | 1.361 |  0.064 |
> | Llama-2-7B       | Birth years | 0.006 | 0.112 | 0.842 |  0.215 | 0.387 | 1.106 |  0.156 |
> | Llama-2-7B       | Populations | 0.000 | 0.195 | 1.118 |  0.147 | 0.314 | 1.231 |  0.033 |

---

### Author Response · Authors · 2025-12-04
**Final Comment to AC**

We thank the reviewers and AC for their time. In addition to our replies to the reviewers, we provide this short summary :


* Reviewer jbmZ reconsidered their view after the rebuttal, explicitly calling the non-linear relationship between compressed representations and a linear number system **“profound”** and raising their score from 4 to 6.

* Reviewer 9VTM is already positive on contribution and methodology; we added targeted behavioral checks and cross-model results directly addressing the robustness and performance questions.

* Reviewer W9st raised concerns about relying on a single prompt, but the suggested equation-style setup was already present in the original paper (with k=1–4); we clarified this and strengthened the analysis by adding the k=0 case, and by expanding model coverage in both Experiments 1 and 2.

---

### Meta-Review · Area_Chair_x7a9 · 2025-12-24

**Summary:**

The paper investigates the number representations of LLMs. The main claim is that LLMs encode numbers logarithmically.

The proposed problem is interesting and impactful, and all the reviewers felt this way.

However, the methodology and results which support the claim could be stronger. Several concerns were pointed out by the reviewers, such as the real-world experiments using birth years which may not be as associated with quantity. The contrast between PCA and PLS also raise questions about the claim. In fact, it may just be the case that LLMs encode numbers both logarithmically and linearly (in different directions of the embedded space), which would be consistent with variations seen in the results. I also believe that the authors could do a different PLS probe, where they fit the representations to the logarithm of the number. This seems like a more direct test of finding whether logarithmic representations exist. Along similar lines, in addition to the scaling rate index, looking at quality of fit to a logarithmic scale would be a more direct validation of logarithmic encoding, instead of only measuring the degree of concavity/convexity.

I also think tokenization artifacts are not ruled out, as raised by one of the reviewers. It is good that authors did a control experiment, however a random sequence of letters is used in the control. Since there is no reason for the LLMs to collate information across random letters, this does not seem like a good control. Using related tokens would be better, since it would test if the observed difference is due to the number of tokens which are being aggregated.

Overall, the paper has merits and could possibly be accepted if there was enough room, but I'd say it is below the bar right now.

**Reviewer Concerns:**

I think concerns around PCA vs PLS, tokenization artifacts, and real-world experiments are still outstanding.

**Reviewer Scores:**

Reviewer jbmZ said they will increase their score, though they did not elaborate any further or say whether their original concerns have been addressed.

---

### Decision · Program_Chairs · 2026-01-26

Reject